# A Study of the Toxic Effect of Plant Extracts against *Philaenus spumarius* (Hemiptera: Aphrophoridae)

**DOI:** 10.3390/insects14120939

**Published:** 2023-12-11

**Authors:** Domenico Rongai, Erica Cesari, Sabrina Bertin

**Affiliations:** 1CREA Research Centre for Engineering and Agro-Food Processing, via Nazionale 38, 65012 Cepagatti, Italy; 2CREA Research Centre for Plant Protection and Certification, via C.G Bertero, 22, 00156 Rome, Italy; erica.cesari@crea.gov.it (E.C.);

**Keywords:** natural compounds, insecticidal activity, *Xylella fastidiosa*, *Taxus baccata*, *Salvia guaranitica*, *Capsicum annuum*

## Abstract

**Simple Summary:**

Olive quick decline syndrome (OQDS) caused by *Xylella fastidiosa* subsp. *pauca* is responsible for extensive desiccation and die-off of olive trees in Apulia (Italy). Current OQDS management strategies include insecticide applications to control the insect vector *Philaenus spumarius*. In addition to the mandatory phytosanitary measures, the demand for new strategies compatible with integrated pest management is increasing. In this study, laboratory biological assays were performed to assess the potential toxic effect of natural compounds against *P. spumarius* adults. Two vegetal formulations were tested: Form A and Form B, the former resulting to be toxic against *P. spumarius*. Indeed, the average rate of spittlebug mortality obtained with Form A ranged from 32.6 to 100% one hour after treatment. In field trials, the number of *P. spumarius* specimens captured on olive plants treated with Form A was significantly lower than on untreated plants. No symptoms of phytotoxicity were recorded on olive trees treated with Form A. This suggests that the tested vegetal formulation is a valid alternative to chemical insecticides for the control of the main vector of *X. fastidiosa* and could be integrated into a sustainable management system for OQDS.

**Abstract:**

The meadow spittlebug *Philaenus spumarius* (Hemiptera: Aphrophoridae) is distributed in several habitats worldwide and has been recently recognized as the main vector of *Xylella fastidiosa* subsp. *pauca*. This bacterium has been associated with olive quick decline syndrome (OQDS) in the Salento Peninsula (Italy) and is responsible for extensive desiccation and die-off of olive trees. Current OQDS management strategies include the control of *P. spumarius* populations, mainly through the removal of weed hosts and insecticide treatments. In addition to the mandatory phytosanitary measures, the demand for new strategies compatible with integrated pest management is increasing. In this study, laboratory biological assays were performed to assess the potential toxic effect of vegetal formulations against *P. spumarius* adults. Two formulations were tested at different concentrations: Form A, an emulsion of 10% hot pepper-infused oil (*Capsicum annuum* subspecies *Cayenna* in olive oil) and Arabic gum in an aqueous solution of extracts of *Salvia guaranitica*, and Form B, an aqueous solution of extracts of *Taxus baccata*. Both Form A and B showed to be toxic against *P. spumarius* compared to the water control. The mean percentage of spittlebug mortality obtained with Form A one hour after treatments was dose–dependent; the lethal dose values were 0.13% (LD25), 0.36% (LD50), and 0.85% (LD75). At the same time, no significant differences in mortality rate were observed between the 0.75% treatment and the treatments with deltamethrin (about 90%). The mean percentage of spittlebug mortality obtained with Form B ranged from 21% to 53% one hour after treatment, but these values were significantly lower than those obtained with deltamethrin. The effectiveness of Form A on the *P. spumarius* population was also evaluated in the field. The averages of captures in the three experimental blocks were 1.8/trap for treated and 7.7/trap for untreated plots, and the spittlebug populations significantly decreased after treatments. Based on these results and the literature data, we hypothesize that the effectiveness of Form A is the result of the synergistic effect of all its components. No symptoms of phytotoxicity were recorded on olive trees treated with Form A, and the number of *P. spumarius* specimens collected on these plants was much lower than on untreated plants. These results suggest the potential use of Form A in the protection of olive trees. This vegetal formulation can thus be considered as a valid alternative to chemical insecticides for the control of the main vector of *X. fastidiosa* and could be integrated into a sustainable management system for OQDS.

## 1. Introduction

The meadow spittlebug *Philaenus spumarius* L. (Hemiptera: Aphrophoridae) is a very common species distributed worldwide in a variety of habitats such as meadows, abandoned fields, waste ground, roadsides, streamsides, and cultivated fields [1]. The direct damages of *P. spumarius* to cultivated plants have been neglected for a long time until the outbreaks of the olive (*Olea europaea* L.) quick decline syndrome (OQDS) in the Salento Peninsula, Italy, occurred [2]. This olive disease is associated with the *Xylella fastidiosa* subsp *pauca* strain “De Donno” ST53, a systemic bacterium that prevents sap flow by colonizing the xylem vessels and causes extensive desiccation and tree die-off. The disease was rapidly responsible for severe losses and drastic landscape changes and was reported on about 23,000 ha only one year after the first report in 2013 [3]. *Philaenus spumarius* has been recognized as the main vector of this bacterium [4,5]. After living on herbaceous dicotyledonous, as nymphs, spittlebug adults preferentially move to woody plants when the hot temperatures and scarce rains dry out the ground vegetation. While feeding from infected olive trees, they can acquire and transmit *X. fastidiosa* [6,7]. The inoculation of the bacterial cells occurs a few minutes after the insertion of spittlebug stylets into the host plant and contact with xylem vessels [8]. The density of the spittlebug nymphs in the olive groves can be very high, reaching 10–40 individuals per square meter, and this can explain the great potential of bacterium transmission once the adults reach the canopies [6,9]. Other less efficient vectors are the spittlebug species *Philaenus italosignus*, the other *Philaenus* species present in Italy but at lower abundance, and *Neophilaenus campestris*, which can both potentially transmit the bacterium but rarely feed on olive trees [10].

Since there are no direct curative methods for the infected plants, OQDS control management relies on the integration of several strategies, including approaches for reducing the cell density of the bacterium within the xylem tissue, the maintenance of soil fertility, regular pruning, and the use of tolerant or resistant olive cultivars [3]. In this frame, it is essential to implement both monitoring and control strategies against *P. spumarius* [11]. The management of spittlebug populations mainly relies on weed removal and mandatory insecticide treatments (D.G.R. n. 1866 of 12 December 2022 and D.G.R. n. 570 of 26 April 2023) to reduce the nymphs that do not harbor the bacterium yet. A recent investigation on local and landscape factors determining the density of *P. spumarius* in the Abruzzo region found that the meadow spittlebug is negatively associated with the presence of vineyards in the landscape. Furthermore, soil management and pesticide applications did not limit the vector, likely because both interventions were not timed based on the biology and ecology of *P. spumarius* [12]. Mechanical weeding carried out during the winter on Salento farms reduced the number of spittlebug nymphs by 70% and, consequently, the number of adults and the spread of *X. fastidiosa* infections [13]. Also, treatments on the foliage in spring to kill *P. spumarius* adults that have moved to olive trees are very important [14]. Unfortunately, these strategies are not sufficient, and *X. fastidiosa* continues to spread at a rate of approximately 10 km each year [15].

Among the tested insecticides, neonicotinoids (imidacloprid) and pyrethroids (deltamethrin and lambda-cyhalothrin) showed a high rate of spittlebug mortality, while the action of the dimethoate-based formulations was weak and slow over time [16,17]. These compounds are listed in the mandatory phytosanitary treatments in the frame of the OQDS management strategy. However, under the current paradigm of severe restrictions on the use of chemical insecticides, it is crucial to search for alternative strategies compatible with integrated management practices (IPMs). Different alternative approaches have been investigated, such as the direct application of some entomopathogenic nematodes and entomopathogenic fungi against nymphs and adults, respectively [18,19,20]; the survey for natural enemies, both egg parasitoids and generalist predators [11]; and the exploitation of semiochemicals [21]. Recommended agronomic measures to reduce *P. spumarius* populations include tillage to eliminate weeds [22] and cover crop management [23]. *Philaenus spumarius* depends a lot on the herbaceous cover of the ground; for example, the eggs are laid among the vegetation in autumn and the larvae and young adults feed on fresh grass in spring before attacking the olive tree [7,24]. Other defense strategies are based on the fact that vibrations could play an important role in the short-distance communication of insects [25]. Therefore, through the interference of the vibrational communication of insects, it is potentially possible to manipulate mating, permanence on hosts, oviposition, and feeding [26,27]. By means of vibrations, the spittlebug could also communicate with its peers to perceive the environment and detect natural enemies. By studying more deeply the role of vibrations in host search, we could potentially repel spittlebugs from olive trees by reducing interactions with the plant and thus limit the spread of *X. fastidiosa* [28]. However, these strategies alone are not capable of controlling spittlebugs but must be combined with other integrated management tools, such as adequate cultivation practices and the use of biopesticides.

In the context of the OQDS-IPM, natural compounds used as biopesticides can contribute to the sustainable control of *P. spumarius* [29]. The aim of this study is to assess the effectiveness of formulations made up of extracts of *Taxus baccata* L., *Salvia guaranitica* St. Hil., *Capsicum annuum* L., and olive oil against meadow spittlebug adults in both laboratory bioassays and field experiments.

## 2. Materials and Methods

### 2.1. Insect Collection and Rearing

*P. spumarius* adults were collected by means of a sweeping net on wild grass in urban parks in Rome (Italy) in April–May and September–October 2020–2021. The specimens were caged in Bug Dorm Insect Rearing Tents (W75 × D75 × H115 cm; mesh size: 44 × 32 µm) on broad bean plants (*Vicia faba* L.) until laboratory experiments. The mesh cages were maintained in a screenhouse without temperature or humidity control to simulate field conditions, and the wilting plants were routinely replaced. A subset of the specimens collected in autumn 2020 were left to overwinter onto the broad beans that were kept until the emergence of the first-instar nymphs in spring, and new generation adults were used in the experiments carried out in 2021.

### 2.2. Preparation of Plant Extracts

Extracts of *T. baccata* (TBE) and *S. guaranitica* (SGE) were selected based on their antifungal activity that was previously assessed [30]. We hypothesized that the same extracts could also work against insects because many substances contained in them had insecticidal activity. These extracts were obtained as follows. The leaves were cut and added to a solvent composed of 80% bidistilled water (Milli-Q-System, Millipore, Bedford, UK) and 20% ethanol (analytical grade RPE, Carlo Erba Reagents, Milan, Italy); then, the mixture was sonicated for 15 min and stirred at 40 °C overnight. After ethanol evaporation, the extract was centrifuged, and the supernatant was filtered through a 0.45 μm PTFE filter. The SGE and TBE were stored at −20 °C.

Hot pepper-infused oil (HPIO) was obtained by macerating small pieces of berries and seeds of *C. annuum* subspecies Cayenna in olive oil in the proportion of 1/3 (*w*/*v*) for 10 days.

### 2.3. Toxicity Bioassays

Two formulations, named Form A and B, were tested in laboratory biological assays at different concentrations. Form A is an emulsion of 10% HPIO in Arabic gum added at increasing concentrations of 0.25, 0.50, 0.75, 1.50, and 2% to an aqueous solution of SGE at 0.6%. Form B is an aqueous solution of TBE that was used at increasing concentrations of 0.75, 1.5, 3, and 6%. A further bioassay was performed with 12 mg/L deltamethrin (Decis® Jet, Bayer, Milano, Italy) as a positive control.

For each formulation, bioassays based on a modified version of the dip test [31] were performed to test the toxic effect on *P. spumarius* adults. In these bioassays, ten specimens were immersed for 15 s into the solution in a 250 mL beaker after being shortly chilled to manipulate them. The insects were then immediately transferred into a Petri dish (9 cm in diameter) containing a broad bean leaf sprayed with the same solution. Four replicates of the bioassay and a control replicate consisting of ten specimens immersed in deionized water were performed per test, and each test was repeated twice. Petri dishes were then placed at 22–25 °C for a 16:8 h photoperiod. The percentage of spittlebug mortality was recorded 1, 18, 24, and 48 h after the treatment. The percentage of mortality was calculated as % mortality = 100 × (mortality of sample–mortality of control)/mortality of sample. For Form A, the mean lethal doses LD25, LD50, and LD75 were estimated by the Probit Analysis Program one hour after the treatment. 

### 2.4. Field Experiments 

The efficacy of Form A against *P. spumarius* adults was evaluated in an olive orchard located in Città Sant’Angelo (PE) (latitude 42.521127 N, longitude14.115878 E) in 2023 during the growing season. The orchard was implanted with olive trees cv. Gentile di Chieti at 5 × 6 m from each other. The experimental plan was set up as randomized blocks of treated and untreated (negative control) plots. Three block replicates each represented by a treated and an untreated plot were used, for a total of six plots of about 480 m^2^. Two treatments with Form A at 1% concentration were performed on 30 June and 30 July 2023. The application volume was 1.500 L/ha, and the solution was sprayed using a farm’s atomizer with a pressure of 5 bar on both olive canopy and ground vegetation. No insecticides were applied in the experimental olive groves during the field trial.

The presence of *P. spumarius* was monitored by yellow sticky traps from 9 June to 4 September 2023. Two traps per plot were placed in the lower part of the canopy about 1.5 m above the ground and were replaced every ten days. The spittlebug specimens were counted on each side of the traps.

### 2.5. Statistical Analysis

The data of mortality were analyzed by a one-way ANOVA with means separation by Fisher’s protected LSD test at α = 0.05. The data were arcsine-transformed prior to analysis to correct them for heterogeneity of variance. The probit analysis (EPA Probit Analysis Program version 1.5) was used to estimate the lethal dose for 25, 50, and 75% of populations (LC25, LC50, and LC75). SigmaPlot V10 (Systat Software Inc., San Jose, CA, USA) and Sigma Stat V3.5 (Systat Software Inc., San Jose, CA, USA) were used to obtain the graphs.

## 3. Results

### 3.1. Toxicity Bioassays

In the bioassays performed with Form A, all the tested concentrations caused a mean percentage of mortality of *P. spumarius* that was significantly higher (F = 22.3, (DF = 1,2), *p* < 0.05) than the control just 1 h after treatment. At the same time, no significant differences in rate mortality were observed between the treatments with 0.75% Form A and deltamethrin, and both were about 90% (Figure 1). 

After 18 h, the mortality of the spittlebugs was 41.4, 66.7, 100, 100, and 100% for the concentrations of 0.25, 0.50, 0.75, 1.5, and 2%, respectively. A 100% mortality was also recorded for deltamethrin. Data were significantly higher (F = 25.4, (DF = 1,5), *p* < 0.05) compared to the untreated control where no spittlebugs died. After 24 and 48 h, the mortality of the spittlebugs did not further change in the untreated control for the concentrations of 0.25 and 0.50%.

The mean percentage of mortality obtained with Form A at 1 h after treatments was dose–dependent and increased from 32.6% of the 0.25% concentration to 100% of the 1.5 and 2% concentrations. The corresponding lethal dose values were 0.13% (LD25), 0.36% (LD50), and 0.85% (LD75) (Figure 2). 

Form B used at growing concentrations (0.75–6%) induced a mean percentage of mortality of *P. spumarius* adults that ranged from 21% to 53% 1 h after treatment (F = 19.5, (DF = 1,13), *p* < 0.05). No mortality was observed in the untreated control at the same time. However, the percentage of mortality of deltamethrin was higher than all Form B concentrations (Figure 3). 

### 3.2. Field Experiments 

No significant differences (F = 0.49, DF = (1,10), *p* = 0.49) for block 1 and significant differences (F = 8.01, DF = (1,10), *p* = 0.02) (F = 7.47, DF = (1,10), *p* = 0.02) for blocks 2 and 3 in the number of *P. spumarius* adults captured in treated and untreated plots were detected before Form A applications (Figure 4). The differences (F = 71.65, DF = (1,22), *p* < 0.001) (F = 20.78, DF = (1,22), *p* < 0.001) (F = 12.47, DF = (1,22), *p* < 0.001) between the average of treated and untreated plots were relatively small before the application; then, they dramatically changed after. These differences were observed until late August/early September when the spittlebug population started to decrease in the control orchard. The averages of the captures in the three experimental blocks were 1.8/trap for treated and 7.7/trap for untreated plots (Figure 4).

## 4. Discussion

According to some authors [32,33], the presence of vegetation among the olive groves and the increasingly warm climate favor the development of *P. spumarius* and, therefore, the diffusion and establishment of *X. fastidiosa* in the Apulian olive agroecosystem. The removal of the soil vegetation of olive crops, by tillage, performed in winter and spring significantly reduced the presence of both *P. spumarius* nymphs and *N. campestris* on olive trees [6]. The tillage performed in winter only reduced the *N. campestris* population but not the *P. spumarius* population when compared to the untreated control plots [17]. To overcome this problem, mulching could be a solution, just as the vibration technique can interfere with the behavior of spittlebugs. The latter, in fact, can disturb mating, oviposition, and feeding [28]. In any case, this technique is still the object of research, and practical applications are not yet available to farmers. All this, however, should be combined with other integrated management tools, such as adequate agronomic practices and the use of biopesticides. Regarding the latter, our tests showed that both Form A and B can cause mortality in *P. spumarius* adults. In the laboratory bioassays, Form A was more effective than B, and the 0.75–2% Form A concentrations showed an insecticidal activity that was not significantly lower than deltamethrin. In addition, at high concentrations, the insecticidal activity of Form A was even significantly higher than deltamethrin. This rate obtained with deltamethrin is consistent with Dongiovanni [16], who reported that the mortality of *P. spumarius* adults treated with synthetic insecticides (deltamethrin, lambda-cyhalothrin, and imidacloprid) ranged from 76.7 to 100% three days after treatment. Field experiments seemed to confirm a certain effectiveness of Form A on wild *P. spumarius* populations, as suggested by the observed significant reduction in captures with yellow sticky traps in the treated blocks. From the lethal dose data (LD25, LD50, and LD75), it can be deduced that to reduce the population density of *P. spumarius* in the field by 50%, a treatment volume of 800 L ha^−1^ containing 4.8 kg ha^−1^ extract of *S. guaranitica* (SGE) and 2.9 L of hot pepper-infused oil (HPIO) is required. 

The toxic activity of Form A is likely due to the three-phase products (SGE, berry pieces of *C. annuum* subspecies Cayenna, and olive oil) combined in the solution. The SGE is rich in bioactive compounds, such as the flavonoid cirsiliol and ethyl ester caffeic acid [34], which are already known to have toxic effects against insects. For example, the extract of *Teucrium zanonii* Pamp., which is rich in flavonoids, such as cirsiliol, luteolin, and chrysoeriol, showed an insecticidal action against *Phloeotribus oleae* Latreille [35], while the caffeic acid was toxic against *Helicoverpa armigera* Hübner by inhibiting the gut proteases [36]. Furthermore, it has been found that the *T. zanonii* extracts with the highest insecticidal activity are those obtained using water as a solvent [35]. Other authors believe that this greater activity is due to the ability of water to extract substances, such as diterpenoids and flavonoids [37]. These results agree with our study, which used 80% water as the extraction solvent. Also, *C. annum* leaf extracts showed toxic activity against insects. When used at a concentration of 10%, the extracts caused a high mortality in *Tribolium confusim* adults [38]. Moreover, the *C. annum* berries that are rich in alkaloids, flavonoids, anthocyanins, tannins, and saponins have a high level of insecticidal activity [39]. This activity was mostly ascribed to the presence of the terpenoids eugenol and pulegone [14]. Eugenol has been found to be toxic against the corn weevil (*Sitophilus zeamaisand*), the red flour beetle (*Tribolium castaneum*), the corn borer (*Prostephanus truncatus*, Horn), and the American corn beetle (*Periplaneta americana*) [40]. Pulegone has shown toxicity toward the German cockroach (*Blattella germanica*), houseflies, storage pests, and *Aedes aegypti* mosquitos [41,42]. Olive oil is also known to be potentially harmful to insects due to the obstruction of the spiracles [43]. However, the extracts contained in Form A, when used in the field alone have rather low insecticidal properties. Also, *S. guaranitica* alone has a very weak or even no insecticidal effect, as reported by [44], who observed that the essential oil of *S. guaranitica* showed no insecticidal activity. Even the *C. annuum* extract tested alone showed an overall weak insecticidal activity, as it was effective against *Aphis gossypii* but not against *Ectropis obliqua* [45]. Field treatments with *C. annuum* extract have recorded a very low mortality rate (19.48%) of white mango scal (*Aulacaspis tubercularis*) [46]. Finally, vegetable oils, although harmful to insects, seem to be not sufficient to cause mortality. Indeed, oil treatments against juveniles of *Philaenus spumarius* and *Neophilaenus campestris* (both able to transmit *X. fastidiosa*) have shown very low mortality [47]. Vegetable oil used against the California red scale (*Aonidiella aurantia*) produces a low mortality of approximately 46.0% [48].

Basied on these results and the literature data, we hypothesize that the effectiveness of Form A is due to the synergistic effect of all its components. When the oil is used in combination with the extract of *S. guaranitica* and the berries of *C. annuum* subspecies *Cayenne*, it is much more effective, as it gives the entire formulation a high viscosity (to be more persistent and less toxic to plants) and a high wetting power (to encourage the formation of a film around the insect). In laboratory tests, following 18, 24, and 48 h of treatment, the mortality percentage of the spittlebugs remained almost unchanged for the untreated control and the concentrations 0.25% and 0.50%. This may suggest that Form A has good efficacy at a concentration equal to or greater than 0.75%. Furthermore, from the results obtained, we can hypothesize that it acts during the first minutes of treatment and that the effectiveness weakens in the following hours. Based on laboratory tests and what was written above, we felt that 1% was the best concentration for field experiments. The dates of the treatments were chosen as they correspond (for the area) to the period of the maximum presence of the spittlebug.

Further studies will be carried out to verify the effectiveness of Form A on other phytophagous insects in olive trees. No symptoms of phytotoxicity were recorded on olive trees treated with Form A, suggesting its possible use in the protection of olive trees.

To conclude, the results of the present work indicate that this vegetal formulation could be a valid alternative to chemical insecticides for the control of the main vector of *X. fastidiosa* and could be integrated into a sustainable management system for OQDS. However, to confirm the validity of the results, further efficacy tests will be necessary. It will also be necessary to improve the formulation technology of the product to obtain the best distribution on olive groves.

The substitution of synthetic pesticides with natural plant molecules significantly reduces the amount of chemicals in the environment and undesired effects, such as the contamination of the food chain, and this is also an important target for the agricultural policies of the European Union.

## Figures and Tables

**Figure 1 insects-14-00939-f001:**
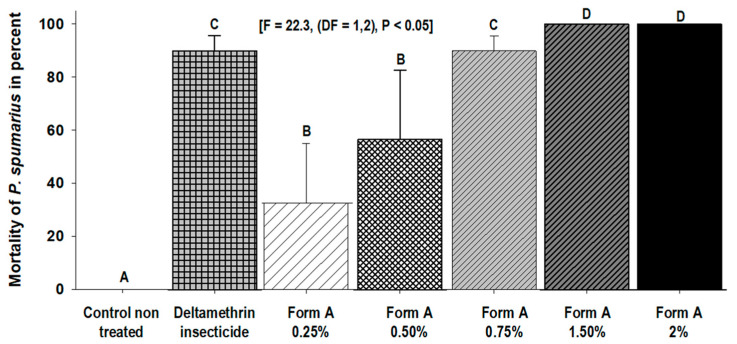
Mean percentage (±SD) of mortality of *P. spumarius* adults recorded 1 h after treatment with different doses (0.25–2%) of Form A and deltamethrin. Values with different letters are statistically different (LSD test, α = 0.05).

**Figure 2 insects-14-00939-f002:**
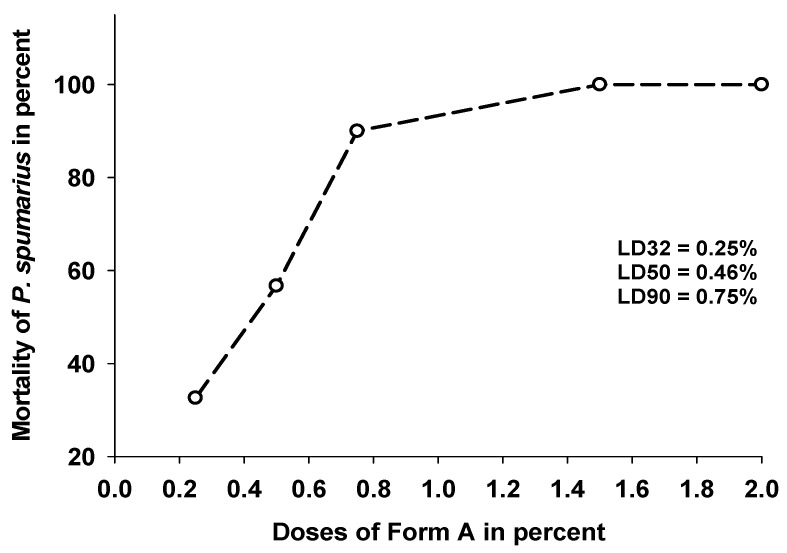
Lethal doses of different doses (0.25–2%) of Form A on *P. spumarius* 1 h after treatment.

**Figure 3 insects-14-00939-f003:**
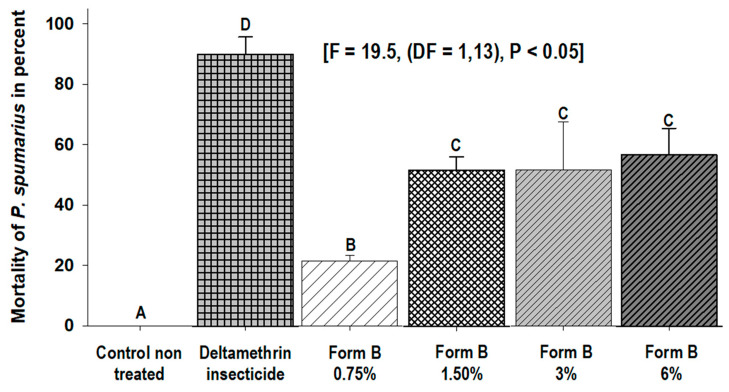
Mean percentage (±SD) of mortality of *P. spumarius* adults recorded 1 h after treatment with different doses (0.75–6%) of Form B and deltamethrin. Values with different letters are statistically different (LSD test, α = 0.05).

**Figure 4 insects-14-00939-f004:**
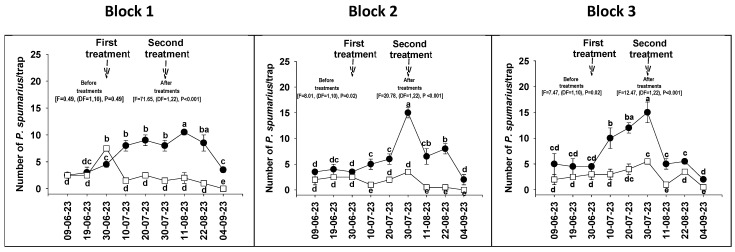
Number (±SD) of *P. spumarius* adults captured from 9 June 2023 to 4 September 2023 in the three experimental blocks after two treatments with Form A at a 1% concentration. Untreated plots ●⸺●; treated plots □⸺□. Values with different letters are statistically different (LSD test, α = 0.05).

## Data Availability

All the data supporting the results of the study are already available in the Results section and in the Figures.

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
