# Peer review of "A Study of the Toxic Effect of Plant Extracts against Philaenus spumarius (Hemiptera: Aphrophoridae)"

_insects, 2023, doi:10.3390/insects14120939_

Round 1
Reviewer 1 Report
Comments and Suggestions for Authors
The aim of the study was to estimate toxic effects of two formulations of plant extracts (as alternatives to chemical insecticides) on the insect vector of the bacterium causing the olive quick decline syndrome, the meadow spittlebug Philaenus spumarius. With this aim the authors conducted laboratory and field experiments. The experiments were well planned and performed; the data were correctly treated and clearly presented; the conclusions made by the authors are supported by the data. The results of the study can be used for the development of a new effective and environmental friendly method for the control of a serious insect pest. Thus, I think that the manuscript can be published, although a number of minor changes and corrections are required (see below).
Line 10: Delete “it” in the end of the line.
Line 12: Delete the dot after “Philaenus”.
Lines 131 and 133: Do not capitalize “autumn” and “spring’.
Lines 138 and 155: Either delete “by” or insert the name of an author.
Figure 2: Linear regression equation and corresponding line are rather far from evidently non-linear “dose–effect” response. Thus, I would suggest deleting it. Lethal doses estimated by the probit analysis are enough to describe the observed experimental effect.
Lines 200-201: What are you mean “for each group”? I see only one group of data on this figure.
Line202: Insert the name of an author after “to”.
Line 218: Put a dot after “deltamethrin”.
Lines 218-219: What are you mean by “for each group”? I see only one group of data on this figure.
Figure 4: If possible, it would be good to conduct some test for multiple pairwise comparing (Fisher’s LSD or Tukey’s HSD) and to show the results by letters, as in Figs. 1 and 3.
Lines 240-242: This sentence seems to be incomplete (a verb after “could” is absent).
Line 247: First, the commonly used term is not “insecticidial” but insecticidal activity.
Second, as seen in Fig. 1, at high concentrations, the insecticidal activity of the form A was not only “not significantly lower” but even significantly higher than that of deltamethrin.
Line 261, 275 etc.: Replace “insecticidial” by “insecticidal” everywhere.
Line 272: Please, explain what insects are the “storage parasites”. Possibly, you mean “storage pests”?
Line 281: Delete “it”.
Author Response
Dear Editor,
I am pleased to have a chance to resubmit my manuscript to the Insects.
Answers to the manuscript comments:
Reviewer #1:
The aim of the study was to estimate toxic effects of two formulations of plant extracts (as alternatives to chemical insecticides) on the insect vector of the bacterium causing the olive quick decline syndrome, the meadow spittlebug Philaenus spumarius. With this aim the authors conducted laboratory and field experiments. The experiments were well planned and performed; the data were correctly treated and clearly presented; the conclusions made by the authors are supported by the data. The results of the study can be used for the development of a new effective and environmental friendly method for the control of a serious insect pest. Thus, I think that the manuscript can be published, although a number of minor changes and corrections are required (see below).
Line 10: Delete “it” in the end of the line. Ok, we deleted it
Line 12: Delete the dot after “Philaenus”. Ok, we deleted it
Lines 131 and 133: Do not capitalize “autumn” and “spring’. Ok, we changed it
Lines 138 and 155: Either delete “by” or insert the name of an author. Ok, we deleted it
Figure 2: Linear regression equation and corresponding line are rather far from evidently non-linear “dose–effect” response. Thus, I would suggest deleting it. Lethal doses estimated by the probit analysis are enough to describe the observed experimental effect. Ok, we deleted it
Lines 200-201: What are you mean “for each group”? I see only one group of data on this figure. Ok, we changed it
Line202: Insert the name of an author after “to”. Ok, we added it
Line 218: Put a dot after “deltamethrin”. Ok, we added it
Lines 218-219: What are you mean by “for each group”? I see only one group of data on this figure. Ok, we changed it
Figure 4: If possible, it would be good to conduct some test for multiple pairwise comparing (Fisher’s LSD or Tukey’s HSD) and to show the results by letters, as in Figs. 1 and 3. The data were also analyzed with Fisher's LSD. Ok we added letters.
Lines 240-242: This sentence seems to be incomplete (a verb after “could” is absent). Ok, we changed it
Line 247: First, the commonly used term is not “insecticidial” but insecticidal activity. Ok
Second, as seen in Fig. 1, at high concentrations, the insecticidal activity of the form A was not only “not significantly lower” but even significantly higher than that of deltamethrin. Ok, we added it
Line 261, 275 etc.: Replace “insecticidial” by “insecticidal” everywhere. Ok, we changed them
Line 272: Please, explain what insects are the “storage parasites”. Possibly, you mean “storage pests”? Ok, we changed it
Line 281: Delete “it”. Ok, we deleted it
With best regards,
Domenico Rongai
Reviewer 2 Report
Comments and Suggestions for Authors
The article is interesting in that it deals with new plant-derived substances that have effects in reducing the fitness of the spittlebug, Philaenus spumarius, main vector of Xylella fastidiosa in southern Italy. The research concept is clear and the experimental design is correct. However, I have a couple of major concerns that deserve careful attention by the authors.
Statistics are not convincing. In the text, I often found the terms "significant" and "not significant"differences. However, it is important to report the actual and precise statistical values of ANOVA (F, df and p, in the main text and/or in the graph captions), whenever statistics are mentioned. The posthoc test is meaningful only if the omnibus test is significant. Another point is that the arcsine transformation must also assessed. Transformed data do not mean automatically homogeneous data. Also, distribution normality must be checked. Then, which kind of ANOVA did you use? I guess it was a ONE-WAY ANOVA, but it must be clearly written. Another important thing: where are the results at 8, 16, 24 and 48 hours? You mentioned them in materials and methods but then they did not appear in results. I think that it could be useful to see these results especially in the case of less effective concentrations. Otherwise, they must be removed from the Materials and Methods chapter.
Another important statistical issue is the total absence of statistics in the field experiments. They are not described in the Materials and Methods while in the Results it is reported on lines 222-225 that no significant differences were observed before the application but later they were found. Again, it is necessary to provide statistical numbers.
In addition, in Figure 2 a R2 value is reported . The R2 value is a coefficient of linear regression. However, a linear regression analysis is not described in the Materials and Methods. Also consider that the curve shown in the figure is not linear and other should be preferred.
In conclusion, I strongly recommend seeiking advice from some statisticians.
The second major point regards the discussion. There are many speculations, especially in the final part, that should be rewritten as hypotheses that deserve further research to confirm what here observed.
Lines 286-290: This part is mere speculation. It can be argued that this is a possibility, but it is not proven. Rewrite this sentence by omitting assertions and only formulating hypotheses.
Lines 295-297: The product is promising but probably not yet ready for large scale applications. I think that it would be more reasonable to discuss perspectives in terms of further research (to confirm the validity of the results) and next steps towards the commercialization.
Then, I have several minor comments:
The Simple Summary is poorly written: the quality of the English is inferior than to the rest of the article. There are several mistakes also including decimal numbers that are preceded by "." and not ",". We need to say what Form A and B are (the term plant-derived formulations seems more correct to me and does not need the "quotation marks", here and elsewhere in the ms).
The English quality of the abstract is much better, although at lines 43 and 44 there is again the problem with commas preceding decimal numbers. At line 45, I recommend to replace "...the effectiveness of Form A is the result..." with "we speculate that the effectiveness of Form A is likely the result", because this work did not investigate the mechanisms of actions.
Introduction
Lines 74-75: another possible reason is that the other 2 spittlebugs are much less abundant in the enviroment than P. spumarius.
Lines 81-82: I recommend to mention that phytosanitary treatments against the vector are mandatory in Apulia (see: D.G.R. n. 1866 del 12/12/2022 e D.G.R. n. 570 del 26/04/2023)
Line 86: correct: "pesticide applications"
Line 104: replace "semiochemistry signals" with "semiochemicals"
Line 121: I would replace "made up of" with "based on". I would also introduce the definition of "plant derived formulations" at this point of the ms.
Materials and Methods
Line 129-133: how often did you replace the broadbeans? What was the size of the mesh cages? Which was their brand? Did first instars come from field collections or from the rearing?
Line 138: The biological activity previously assessed for the selected extracts concerned fungi. This needs to be clearly mentioned. I would also recommend to mention that you have hypothesized that the same extracts may also work against insects, although it seems a bit far-fetched since the mechanisms of action are not known according to the cited article. Perhaps, do you have some preliminary results or observations that support the hypothesis?
Line 138-139: correct: "as follows"
Line 153: add at the end of the sentence "as a positive control".
Lines 157-158: Please provide more details: how did you manipulate the insects? I know that handling adult spittlebugs is not an easy task. How large was the container?
Line 162: the check times after 1 hour are reported only here, while in the Results they were omitted. Please, see also my major comments.
Line 173: add "(negative control)"
Line 175: why did you choose a concentration of 1% that was not used in the laboratory tests? How did you choose the dates of treatments?
Line 181-182: This is important. Sticky traps do not indicate quantity. This is well known to all scientists working with sticky traps. In fact, there is not strict correlation with the actual population of the target species. However, we can conclude from your experiments that there were fewer captures in the treated plots. I agree that the first hypothesis of the reduction of captures is possibly related to decrease in population but there could be other reasons as well, such as lower mobility or flying/jumping ability of the treated individuals. This would be also a clear reduction in fitness but not in population. This needs to be considered in the discussion.
Lines 203-204: This sentence must be moved to the discussion.
Lines 213-214: Replace the sentence with: "However, the percentage of mortality of deltamethrin was higher than that of all Form B concentrations"
Line 226: delete "the": "until the late August/ early September"
Line 227: rephrase: "...in the control orchard as well".
Discussion
Line 237: Other authors suggest (missing references)
Line 238-240: I do not agree with this statement. Nymphs are not much mobile and they live in the spittle. If you cut their herbs they hardly move, especially to climb trees. In case I am wrong, please, provide the references.
Line 241-242: provide references to the technique of behavioral interference. It is also important to write that this technique is still object of research and practical applications are not yet available to farmers.
Lines 243-244: replace: "agronomic practices"
Lines 248: Here would be the right place to comment on sticky traps and their lack of reliability with respect to actual numbers of individuals. I am confident there is an effect but what we can say is that there are fewer captures. Nobody has checked the actual number of individuals on the vegetation. This would possible by sampling with sweep nets.
Line 272: add: "the mosquito"
Line 280: correct: "19.48"
Line 281 and 284: correct: "vegetable oils"
Comments on the Quality of English Language
I have included my comments on English in previous comments.
Author Response
Dear Editor,
I am pleased to have a chance to resubmit my manuscript to the Insects.
Answers to the manuscript comments:
Reviewer #2:
|
Yes |
Can be improved |
Must be improved |
Not applicable |
|
|
Does the introduction provide sufficient background and include all relevant references? |
( ) |
(x) |
( ) |
( ) |
|
Are all the cited references relevant to the research? |
( ) |
(x) |
( ) |
( ) |
|
Is the research design appropriate? |
(x) |
( ) |
( ) |
( ) |
|
Are the methods adequately described? |
( ) |
( ) |
(x) |
( ) |
|
Are the results clearly presented? |
( ) |
( ) |
(x) |
( ) |
|
Are the conclusions supported by the results? |
(x) |
( ) |
( ) |
( ) |
Comments and Suggestions for Authors
The article is interesting in that it deals with new plant-derived substances that have effects in reducing the fitness of the spittlebug, Philaenus spumarius, main vector of Xylella fastidiosa in southern Italy. The research concept is clear and the experimental design is correct. However, I have a couple of major concerns that deserve careful attention by the authors.
Statistics are not convincing. In the text, I often found the terms "significant" and "not significant"differences. However, it is important to report the actual and precise statistical values of ANOVA (F, df and p, in the main text and/or in the graph captions), whenever statistics are mentioned. Ok, we added them
The posthoc test is meaningful only if the omnibus test is significant. Another point is that the arcsine transformation must also assessed. Transformed data do not mean automatically homogeneous data. Also, distribution normality must be checked. Then, which kind of ANOVA did you use? I guess it was a ONE-WAY ANOVA, but it must be clearly written. Yes we used One-way ANOVA. It has been added in the manuscript.
Another important thing: where are the results at 8, 16, 24 and 48 hours? You mentioned them in materials and methods but then they did not appear in results. I think that it could be useful to see these results especially in the case of less effective concentrations. Otherwise, they must be removed from the Materials and Methods chapter. Ok, we added them in the results chapter
Another important statistical issue is the total absence of statistics in the field experiments. They are not described in the Materials and Methods while in the Results it is reported on lines 222-225 that no significant differences were observed before the application but later they were found. Again, it is necessary to provide statistical numbers. Ok, we've added the statistics on the text and in the graphs.
In addition, in Figure 2 a R2 value is reported . The R2 value is a coefficient of linear regression. However, a linear regression analysis is not described in the Materials and Methods. Also consider that the curve shown in the figure is not linear and other should be preferred. Ok we deleted linear regression. Lethal doses estimated by the probit analysis are enough to describe the observed experimental effect
In conclusion, I strongly recommend seeiking advice from some statisticians.
The second major point regards the discussion. There are many speculations, especially in the final part, that should be rewritten as hypotheses that deserv e further research to confirm what here observed.
Lines 286-290: This part is mere speculation. It can be argued that this is a possibility, but it is not proven. Rewrite this sentence by omitting assertions and only formulating hypotheses. Ok, we rewrited it as a hypothesis
Lines 295-297: The product is promising but probably not yet ready for large scale applications. I think that it would be more reasonable to discuss perspectives in terms of further research (to confirm the validity of the results) and next steps towards the commercialization. Ok, we rewrited it
Then, I have several minor comments:
The Simple Summary is poorly written: the quality of the English is inferior than to the rest of the article. There are several mistakes also including decimal numbers that are preceded by "." and not ",". We need to say what Form A and B are (the term plant-derived formulations seems more correct to me and does not need the "quotation marks", here and elsewhere in the ms).
The English quality of the abstract is much better, although at lines 43 and 44 there is again the problem with commas preceding decimal numbers. At line 45, I recommend to replace "...the effectiveness of Form A is the result..." with "we speculate that the effectiveness of Form A is likely the result", because this work did not investigate the mechanisms of actions.
Introduction
Lines 74-75: another possible reason is that the other 2 spittlebugs are much less abundant in the enviroment than P. spumarius. This is true for P. italosignus but not for N. campestris. The sentence has been modified accordingly.
Lines 81-82: I recommend to mention that phytosanitary treatments against the vector are mandatory in Apulia (see: D.G.R. n. 1866 del 12/12/2022 e D.G.R. n. 570 del 26/04/2023) Ok, we added it
Line 86: correct: "pesticide applications" Ok, we added it
Line 104: replace "semiochemistry signals" with "semiochemicals" Ok, we changed it
Line 121: I would replace "made up of" with "based on". I would also introduce the definition of "plant derived formulations" at this point of the ms.
Materials and Methods
Line 129-133: how often did you replace the broadbeans? This information was added in the text.
What was the size of the mesh cages? Which was their brand? This information was added in the text
Did first instars come from field collections or from the rearing? The first instars mentioned in this paragraph hatched in spring 2021 from the eggs laid in the broadbean’s soil by the adults that we collected from the field during the previous autumn 2020.
Line 138: The biological activity previously assessed for the selected extracts concerned fungi. This needs to be clearly mentioned. I would also recommend to mention that you have hypothesized that the same extracts may also work against insects, although it seems a bit far-fetched since the mechanisms of action are not known according to the cited article. Perhaps, do you have some preliminary results or observations that support the hypothesis? Ok. We specified it in the text
Line 138-139: correct: "as follows" Ok, we added it
Line 153: add at the end of the sentence "as a positive control". Ok, we added it
Lines 157-158: Please provide more details: how did you manipulate the insects? I know that handling adult spittlebugs is not an easy task. How large was the container? These information were added in the text.
Line 162: the check times after 1 hour are reported only here, while in the Results they were omitted. Please, see also my major comments. Ok, we added these informations in the results.
Line 173: add "(negative control)"
Line 175: why did you choose a concentration of 1% that was not used in the laboratory tests? How did you choose the dates of treatments? Based on laboratory tests we felt that 1% was the best concentration for field experiments. As regards the dates of the treatments, they were chosen as they correspond (for the area) to the period of maximum presence of the spittlebug.
Line 181-182: This is important. Sticky traps do not indicate quantity. This is well known to all scientists working with sticky traps. In fact, there is not strict correlation with the actual population of the target species. However, we can conclude from your experiments that there were fewer captures in the treated plots. I agree that the first hypothesis of the reduction of captures is possibly related to decrease in population but there could be other reasons as well, such as lower mobility or flying/jumping ability of the treated individuals. This would be also a clear reduction in fitness but not in population. This needs to be considered in the discussion. The sentence has been modified accordingly
Lines 203-204: This sentence must be moved to the discussion. Ok, we changed it
Lines 213-214: Replace the sentence with: "However, the percentage of mortality of deltamethrin was higher than that of all Form B concentrations" Ok, we changed it
Line 226: delete "the": "until the late August/ early September" Ok, we changed it
Line 227: rephrase: "...in the control orchard as well". Ok, we changed it
Discussion
Line 237: Other authors suggest (missing references) Ok, we added it
Line 238-240: I do not agree with this statement. Nymphs are not much mobile and they live in the spittle. If you cut their herbs they hardly move, especially to climb trees. In case I am wrong, please, provide the references. Ok, we changed it
Line 241-242: provide references to the technique of behavioral interference. Ok, we added it
It is also important to write that this technique is still object of research and practical applications are not27]] yet available to farmers. Ok, we added it
Lines 243-244: replace: "agronomic practices" Ok, we changed it
Lines 248: Here would be the right place to comment on sticky traps and their lack of reliability with respect to actual numbers of individuals. I am confident there is an effect but what we can say is that there are fewer captures. Nobody has checked the actual number of individuals on the vegetation. This would possible by sampling with sweep nets. The sentence has been modified accordingly.
Line 272: add: "the mosquito" Ok, we added it
Line 280: correct: "19.48" Ok, we changed it
Line 281 and 284: correct: "vegetable oils"
With best regards,
Domenico Rongai
Round 2
Reviewer 2 Report
Comments and Suggestions for Authors
The authors have made noteworthy progress in addressing my previous comments, significantly improving the article. I appreciate their efforts. However, a few minor comments still require attention before I can provide my final approval for publication. I encourage the authors to consider these additional suggestions to ensure the manuscript meets the highest standards.
Line 11: add to the end of the first sentence: “in Apulia (Italy).
Line 17: replace “Inded” with “Indeed”.
Line 17: correct: 32.6%
Line 20: replace “plant origin formulation” with vegetal formulation as you have previously defined them at line 16. “” are not necessary.
Line 31: now they are defined as ‘natural compounds’. I recommend to choose only one terminology and use it throughout the manuscript.
Line 40: correct: 21% to 53% one hour after treatment.
Line 43: 1.8/trap 7.75/trap. Please, check the improper use of commas in decimal numbers througout the manuscript. Also, choose to use one or two decimal numbers consistently.
Line 45: Replace "...the effectiveness of Form A is the result..." with "we speculate” or “we hypothesize that the effectiveness of Form A is the result", because this work did not investigate the mechanisms of actions.
Line 48: See my previous comments about the term “plant origin formulation”. Uniform it throughout the manuscript.
Line 67: “after living on herbaceous dicotyledonous as nymphs, spittlebug adults…”
Line 68. “..ground vegetation. While feeding from infected olive trees, they can acquire…”
Line 70: replace “later” with “after”
Line 75: replace “abundance” with “density”
Line 91: Also,
Line 92: to kill P. spumarius adults
Line 93: delete: “ where they feed, acquire the bacterium and transfer it to the nearby tree” because it is a repetition of line 68.
Line 109: “..cover of the ground: for example…”
Line 111-120: the use of vibrations for control is still experimental and the research is in course. Instead, reding this paragraph it seems to be an already available practice. Please, rewrite the full paragraph by using verbs like “could” or “may” and adverbs lke “potentially”.
Results: This is important: The degrees of freedom are never reported. Without this info, it is not possible to make any assessment of significant effect. Therefore, they must be reported.
Line 211: “…spittlebugs did not further change…”
Line 220: from 21% to 53%. It is always preferable to repeat % for both values. Do the same also elsewhere in the text.
Paragraph 3.2 (Field experiments). You reported “no significant differences”, but then the P values are 0.02. These are clearly significant effects. Even if degrees of freedom are not reported, for my experience I can say that values of F = 7 to 8 are likely associated to a significant effect. This means that there was already a difference before the application of the Form A in blocks 2 and 3. In the lines 234-237 and then in the discussion you could say that the differences between the average between treated and untreated plots was relatively small before the application, then they dramatically changed after it.
Line 262: replace “is according to” with “is consistent with that of”
Line 265: You cannot say that “This efficacy was confirmed also in the field experiments”, because we have not numerical data of direct efficacy. All we can say is that “Field experiments seemed to confirm a certain effectiveness of FORM A on wild P. spumarius populations as suggested by the observed significant reduction in captures with yellow sticky traps in the treated blocks”.
Line 304: replace “we could hypotized” with “we could hypothesize”
Line 314: again, reconsider the term “plant origin formulation”.
In the discussion there is not any reference to the results at 8, 16, 24 and 48 hours. If this part is not relevant, then better to remove it from the article. If it has any importance, it must be discussed.
From the previous round of review you responded to my comment but then you did not add the response into the text. I think that this information would be useful for readers.
Line 175: why did you choose a concentration of 1% that was not used in the laboratory tests? How did you choose the dates of treatments? Based on laboratory tests we felt that 1% was the best concentration for field experiments. As regards the dates of the treatments, they were chosen as they correspond (for the area) to the period of maximum presence of the spittlebug.
Comments on the Quality of English Language
In my minor comments I provided several suggestions for English improvements; however, these are not exhaustive. The overall English quality is, on average, rather modest and would benefit from a final proofreading by a native speaker or someone proficient in English.
Author Response
Comments and Suggestions for Authors
The authors have made noteworthy progress in addressing my previous comments, significantly improving the article. I appreciate their efforts. However, a few minor comments still require attention before I can provide my final approval for publication. I encourage the authors to consider these additional suggestions to ensure the manuscript meets the highest standards.
Line 11: add to the end of the first sentence: “in Apulia (Italy). OK we added it
Line 17: replace “Inded” with “Indeed”. OK changed it
Line 17: correct: 32.6% OK changed it
Line 20: replace “plant origin formulation” with vegetal formulation as you have previously defined them at line 16. “” are not necessary. OK changed it
Line 31: now they are defined as ‘natural compounds’. I recommend to choose only one terminology and use it throughout the manuscript. OK changed it
Line 40: correct: 21% to 53% one hour after treatment. OK we added it
Line 43: 1.8/trap 7.75/trap. Please, check the improper use of commas in decimal numbers througout the manuscript. Also, choose to use one or two decimal numbers consistently. OK changed it
Line 45: Replace "...the effectiveness of Form A is the result..." with "we speculate” or “we hypothesize that the effectiveness of Form A is the result", because this work did not investigate the mechanisms of actions. OK changed it
Line 48: See my previous comments about the term “plant origin formulation”. Uniform it throughout the manuscript. OK changed it
Line 67: “after living on herbaceous dicotyledonous as nymphs, spittlebug adults…” OK changed it
Line 68. “..ground vegetation. While feeding from infected olive trees, they can acquire…” OK changed it
Line 70: replace “later” with “after” OK changed it
Line 75: replace “abundance” with “density” OK changed it
Line 91: Also, Ok we added it
Line 92: to kill P. spumarius adults OK changed it
Line 93: delete: “ where they feed, acquire the bacterium and transfer it to the nearby tree” because it is a repetition of line 68. Ok we deleted it
Line 109: “..cover of the ground: for example…” OK we changed it
Line 111-120: the use of vibrations for control is still experimental and the research is in course. Instead, reding this paragraph it seems to be an already available practice. Please, rewrite the full paragraph by using verbs like “could” or “may” and adverbs lke “potentially”. OK we changed them
Results: This is important: The degrees of freedom are never reported. Without this info, it is not possible to make any assessment of significant effect. Therefore, they must be reported. Ok we added them
Line 211: “…spittlebugs did not further change…” I think “the mortality does not further change”
Line 220: from 21% to 53%. It is always preferable to repeat % for both values. Do the same also elsewhere in the text.
Paragraph 3.2 (Field experiments). You reported “no significant differences”, but then the P values are 0.02. These are clearly significant effects. Even if degrees of freedom are not reported, for my experience I can say that values of F = 7 to 8 are likely associated to a significant effect. This means that there was already a difference before the application of the Form A in blocks 2 and 3. In the lines 234-237 and then in the discussion you could say that the differences between the average between treated and untreated plots was relatively small before the application, then they dramatically changed after it. OK we changed it
Line 262: replace “is according to” with “is consistent with that of” OK we changed it
Line 265: You cannot say that “This efficacy was confirmed also in the field experiments”, because we have not numerical data of direct efficacy. All we can say is that “Field experiments seemed to confirm a certain effectiveness of FORM A on wild P. spumarius populations as suggested by the observed significant reduction in captures with yellow sticky traps in the treated blocks”. OK we changed it
Line 304: replace “we could hypotized” with “we could hypothesize” OK we changed it
Line 314: again, reconsider the term “plant origin formulation”. OK changed it
In the discussion there is not any reference to the results at 8, 16, 24 and 48 hours. If this part is not relevant, then better to remove it from the article. If it has any importance, it must be discussed. Ok we added it
From the previous round of review you responded to my comment but then you did not add the response into the text. I think that this information would be useful for readers.
Line 175: why did you choose a concentration of 1% that was not used in the laboratory tests? How did you choose the dates of treatments? on laboratory tests we felt that 1% was the best concentration for field experiments. As regards the dates of the treatments, they were chosen as they correspond (for the area) to the period of maximum presence of the spittlebug. Ok we added it in the discussion.
Comments on the Quality of English Language
In my minor comments I provided several suggestions for English improvements; however, these are not exhaustive. The overall English quality is, on average, rather modest and would benefit from a final proofreading by a native speaker or someone proficient in Eng